# Relationship between Aldehyde Dehydrogenase, PD-L1 and Tumor-Infiltrating Lymphocytes with Pathologic Response and Survival in Breast Cancer

**DOI:** 10.3390/cancers14184418

**Published:** 2022-09-11

**Authors:** Mariana López Flores, Emiliano Honrado Franco, Luis Felipe Sánchez Cousido, Carlos Minguito-Carazo, Oscar Sanz Guadarrama, Laura López González, María Eva Vallejo Pascual, Antonio José Molina de la Torre, Andrés García Palomo, Ana López González

**Affiliations:** 1Medical Oncology Department, University Hospital of León, 24071 León, Spain; 2Pathology Department, University Hospital of León, 24071 León, Spain; 3Cardiology Department, University Hospital of León, 24071 León, Spain; 4General Surgery Department, University Hospital of León, 24071 León, Spain; 5Radiology Department, University Hospital of León, 24071 León, Spain; 6Faculty of Economics and Business, University of León, 24071 León, Spain; 7The Research Group in Gen-Environment and Health Interactions (GIIGAS)/Institute of Biomedicine (IBIOMED), University of León, 24071 León, Spain

**Keywords:** aldehyde dehydrogenase, breast cancer stem cells, breast cancer

## Abstract

**Simple Summary:**

Aldehyde dehydrogenase 1A1 (ALDH1A1) can be used to identify breast cancer stem cells (CSC). Interaction between CSC and the tumor microenvironment might be related to the treatment response, relapse and death. The aim of this retrospective, historical cohort study was to analyze the relationship between ALDH1A1, programmed death ligand 1 (PD-L1) and tumor-infiltrating lymphocytes (TILs), assessed through immunohistochemistry, in triple negative (TN) and human epidermal growth factor receptor 2-positive (HER2+) breast cancer tumors, and its association with clinicopathological characteristics and survival. In this study, tumors with positive ALDH1A1 expression also presented positive PD-L1 expression, higher infiltration of lymphocytes and achieved higher response to treatment. It is crucial to understand the possible implication of these biomarkers on neoadjuvant treatment response in certain subtypes of breast cancer as well as its prognostic role in early and locally advanced breast tumors, as they might be possible targets for promising treatments as immunotherapy.

**Abstract:**

Aldehyde dehydrogenase 1A1 (ALDH1A1) is a cancer stem cell (CSC) marker related to clinical outcomes in breast cancer (BC). The aim of this study was to analyze the relationship between ALDH1A1, programmed death ligand 1 (PD-L1) and tumor-infiltrating lymphocytes (TILs) in triple negative (TN) and human epidermal growth factor receptor 2-positive (HER2+) BC tumors, and its association with clinicopathological characteristics and outcomes. A retrospective, historical cohort study of patients diagnosed with early or locally advanced BC treated with neoadjuvant chemotherapy was conducted. ALDH1A1, PD-L1 expression and TILs were assessed using immunohistochemistry. A total of 75 patients were analyzed (42.7% TN, 57.3% HER2+ tumors). ALDH1A1+ was related to HTILs (*p* = 0.005) and PD-L1+ tumors (*p* = 0.004). ALDH1A1+ tumors presented higher CD3+ (*p* = 0.008), CD4+ (*p* = 0.005), CD8+ (*p* = 0.003) and CD20+ (*p* = 0.006) TILs. ALDH1A1+ (*p* = 0.018), PD-L1+ (*p* = 0.004) and HTILs (*p* < 0.001) were related to smaller tumors. ALDH1A1+ was related to pathologic complete response (pCR) (*p* = 0.048). At the end of the follow-up (54.4 [38.3–87.6] months), 47 patients (62.7%) remained disease-free, and 20 (26.7%) had died. HTILs were related to improved disease-free survival (*p* = 0.027). ALDH1A1+ was related to PD-L1+ and HITLs, that might be related to higher pCR rates with neoadjuvant therapy.

## 1. Introduction

Breast cancer (BC) is the most commonly diagnosed neoplasm worldwide, excluding nonmelanoma skin cancer [1]. Differences in the expression pattern of hormone receptors and the human epidermal growth factor receptor-2 (HER2), and the interaction between tumor cell subpopulations and diverse cells present in the tumor microenvironment, create a highly inter- and intratumoral heterogeneity in these neoplasms [2]. In addition, some subtypes, such as triple negative (TN) and HER2+, exhibit a more aggressive behavior and worst survival outcomes. Unfortunately, despite advances in neoadjuvant and adjuvant treatment, some patients relapse.

A group of initiating cells which exhibit cancer stem cells (CSC) characteristics, such as the ability of self-renewal, differentiation and metastasis, have been suggested as a mechanism for resistance and relapse in BC [3]. Various markers have been proposed for CSC identification, including aldehyde dehydrogenase (ALDH) expression. ALDH is a cytosolic enzyme pertinent to the detoxification of endogenous and exogenous aldehyde substrates through oxidation to carboxylic acid. The ALDH1 isoform is a marker of normal tissue stem cells (SC) and CSC4 [4]. Moreover, ALDH1A1 seems to play a role in the early differentiation of breast cancer stem cells [5]. In addition, in BC, the cytoplasmic expression of ALDH1 has been associated with worse prognosis and lower pathologic complete response (pCR) rates after neoadjuvant treatment [6].

Regarding other sources of intratumoral heterogeneity in particular subtypes, TN and HER2+ have been associated with a more immunogenic tumor microenvironment, with a higher presence of immune cells such as tumor-infiltrating lymphocytes (TILs) [7] and the activation of multiple signaling pathways as programmed death 1/programmed death ligand 1 (PD-1/PD-L1) [8,9]. Studies in BC patients relate a higher presence of TILs and positive PD-L1 expression (PD-L1+) with superior pCR and better survival rates [10].

Considering the unfavorable evolution of these aggressive subtypes, it is crucial to identify factors present in the tumor microenvironment whose interaction might be associated with pathologic response, local relapse or the development of metastasis.

Due to the possible implication of ALDHA1, PD-L1 and the presence of TILs as predictive biomarkers of response and prognosis in BC, it is interesting to explore their relationship. To the best of our knowledge, there are no published data in this respect on BC. Therefore, the aim of the present study was to analyze the relationship between these three biomarkers and its correlation with clinicopathological characteristics, pathologic response after neoadjuvant chemotherapy and survival in BC patients. This might provide tools to define which subgroups of patients would benefit from the administration of immunological therapies in the neoadjuvant treatment of BC.

## 2. Materials and Methods

### 2.1. Study Population

A retrospective, unicentric, historical cohort study of patients diagnosed with early or locally advanced BC treated with neoadjuvant chemotherapy between 2008 and 2018 was conducted. Patients with TN and HER2+ tumors with a histological tumor sample of the diagnosis available who underwent surgery after neoadjuvant treatment were included in this analysis. Metastatic patients and nonevaluative biopsy sample cases were excluded from the analysis.

### 2.2. Objectives

The primary endpoint of the study was to analyze the relationship between ALDH1A1 expression, PD-L1 expression and the presence of TILs in the cohort and its correlation with clinicopathological characteristics in patients. The main clinicopathological findings were age, obesity (body mass index ≥ 30), menopausal status, histological type, histopathological grading, Ki67 and androgen receptor expression, tumor size, lymph node involvement, tumor subtype and pathologic response. Tumor size and lymph node involvement were assessed according to the 8th edition of TNM Classification of Malignant Tumors. Following the completion of neoadjuvant therapy, all patients underwent surgery. Pathologic response was assessed on the resected specimen and all sampled regional lymph nodes. pCR was defined as the absence of residual invasive cancer on hematoxylin and eosin evaluation in the breast and axillary lymph nodes. Secondary outcomes included assessing differences in clinicopathological findings and survival outcomes according to ALDH1A1 expression, PD-L1 expression and TIL infiltration.

### 2.3. Immunohistochemistry Method

The immunohistochemistry method for ALDH1A1 staining was performed in the tumor sample of the diagnostic biopsy, as previously described [11]. The paraffin-embedded tumor tissue samples were cut with a rotary microtome. After deparaffinization and rehydration, they were washed in phosphate buffered saline, and heat-induced antigen retrieval was performed in a pressure cooker for 2 min in EDTA at pH 8. Subsequently, slides were automatically stained and incubated for 1 h with ALDH1A1-specific antibody (Abcam) at a dilution of 1/100. The EnVision + peroxidase complex was used as a visualization system. The product of the antigen antibody reaction was developed with a diaminobenzidine solution and H202. Nuclei were stained with Harris hematoxylin (15 s), and the samples were dehydrated with increasingly concentrated alcohols for final mounting on a permanent medium (Eukitt) (O. Kindler and Co; GMBGH Freiburg, Germany). Afterwards, evaluation of ALDH1A1 immunohistochemistry was performed by scanning the stained slides at medium (20×) and high magnification (100×). The percentage of tumor area with positive cytoplasmic staining was manually estimated by an expert pathologist. Specimens were considered ALDH1A1 positive when the staining was more than 1% in tumor cells.

PD-L1 expression was assessed using Ventana PD-L1 (SP142) assay. Tumor cell and infiltrating immune cell staining were counted. PD-L1 was considered positive when there was more than 1% of positive staining on tumor cells.

TILs where evaluated following international recommendations [12] using 15 as cutoff for high (HTILs) or low (LTILs). Identification of TIL subtypes was performed using the corresponding antibody (CD3, CD4, CD8, CD20). The percentages of TILs corresponding to TILs CD3+, CD4+ and CD8+ were analyzed as quantitative variables and TILs CD20+ as qualitative variables, as previously described [13].

Androgen receptor (AR) expression was assessed through immunohistochemistry and was considered positive when there was more than 1% of positive staining on tumor cells. The histopathological grade of invasive BC was assessed using the Nottingham modification of the Scarff–Bloom–Richardson grading scheme (NSBR). The use of the tissue samples and clinicopathological information of this study was approved by the local medical ethics committee.

### 2.4. Statistical Analysis

Continuous variables were summarized as the mean ± standard deviation (SD) or as the median and interquartile range and compared using the Student’s *t*-test or Mann–Whitney rank sum tests depending on normality. Derangement from the normal distribution was assessed using the Shapiro–Wilk test. Categorical variables were described as percentages and compared using the chi-square or Fisher exact test according to the expected frequency over or below 5. Survival curves for time-to-event analysis were constructed on the basis of all the available follow-up data using Kaplan–Meier estimates. Comparisons between groups were performed using the log rank test. A Cox proportional hazards regression model adjusted by age, tumor size (T stage) and lymph node status (N stage) was performed to evaluate the influence of ALDH1A1 expression, PD-L1 expression and TIL infiltration on disease-free survival (DFS) and overall survival (OS). A *p*-value < 0.05 was considered statistically significant. Statistical analyses were performed using STATA software version 15.1.

## 3. Results

### 3.1. Study Population

During the study period, 104 patients diagnosed with early or locally advanced BC received neoadjuvant chemotherapy. Twenty-two of them with luminal subtype tumors and six with a nonevaluable sample were excluded, while one patient was lost in the follow-up. Finally, 75 patients were analyzed (100% women, mean age 53.6 ± 11.7 years). A total of 32 (42.7%) and 43 (57.3%) were TN and HER2+ tumors, respectively: 21 (28%) had obesity, 32 (42.7%) had a tumor size ≤ 5 cm, and 52 patients (69.3%) had positive lymph nodes. The clinicopathological characteristics of the patients are summarized in Table 1.

Forty (53.3%) patients showed positive ALDH1A1 expression (Figure 1). From them, 18 (24%) tumors presented nuclear ALDH1A1 staining in addition to cytoplasmic expression. Twenty-eight (37.3%) cases showed positive PD-L1 staining (Figure 2). On the other hand, HTIL infiltration was identified in 32 (42.7%) tumors (Figure 3). All the analyzed carcinomas presented the infiltration of TILs CD3+, CD4+ and CD8+. TIL CD8+ was the least common lymphocyte subtype with a median of 5% (3–10%). Median TILs CD3+ [10% (5–20%)] and CD4+ were similar [10% (5–30%)]. Fifty-five (73.3%) tumors presented TIL CD20+. There were no differences in ALDH1A and PD-L1 expression between TN and HER+ tumors, neither in total TIL infiltration nor in the different analyzed TIL subtypes.

Of the entire cohort, and at the end of follow-up (54.4 [38.3–87.6] months), 37 patients (49.73%) achieved pCR after neoadjuvant chemotherapy, 47 patients (62.7%) remained disease-free, and 20 patients (26.7%) had died.

### 3.2. Relationship between ALDH1A1, PD-L1 and TILs

Positive ALDH1A1 carcinomas presented a higher prevalence of HTILs compared to negative ALDH1A1 cases (57.5% vs. 25.7%; *p* = 0.005). This difference was statistically significant in TN tumors (58.8% vs. 20%; *p* = 0.026) but not in HER2+ cases (56.5% vs. 30%; *p* = 0.081). Additionally, positive ALDH1A1 tumors exhibited greater TILs CD3+ (17.5% vs. 10%; *p* = 0.008), CD4+ (20% vs. 10%; *p* = 0.005) and CD8+ (10% vs. 5%; *p* = 0.003) when compared to negative ALDH1A1 cases. Moreover, carcinomas with positive ALDH1A1 staining more frequently showed CD20+ cell infiltration compared to negative ALDH1A1 tumors (85% vs. 60%; *p* = 0.015). However, this difference was only observed in TN (*p* = 0.006) and not in HER2+ carcinomas (*p* = 0.331).

On the other hand, positive ALDH1A1 tumors exhibited a higher frequency of positive PD-L1 staining than negative ALDH1A1 cases (52.5% vs. 20%; *p* = 0.004). In addition, this difference was observed in the TN subtype (58.8% vs. 20%; *p* = 0.026) but not in HER2+ tumors.

In addition, positive PD-L1 cases showed a higher HTIL infiltration than cases without PD-L1 expression (71.4% vs. 25.5%; *p* = 0.001). This difference was also present in the HER2+ subtype (86.7% vs. 21.4%; *p* < 0.001) but not in the TN (53.9% vs. 31.6%; *p* = 0.208). Moreover, positive PD-L1 carcinomas exhibited greater infiltration of all evaluated TIL subtypes in comparison to negative PD-L1 cases: CD3+ (25% vs. 10%; *p* < 0.001), CD4+ (20% vs. 10%; *p* < 0.001) and CD8+ (10% vs. 5%; *p* < 0.001). Additionally, carcinomas with positive PD-L1 staining more frequently showed TIL CD20+ than negative PD-L1 tumors (96.4 vs. 59.6%; *p* < 0.001). All HER2+ tumors with positive PD-L1 staining showed TIL CD20+ infiltration.

Finally, the relationship among the three biomarkers was analyzed by establishing four groups according to positive (PD-L1+) or negative (PD-L1− PD-L1 expression and TIL presence: PD-L1+/HTILs, PD-L1+/LTILs, PD-L1−/HTILs and PD-L1−/LTILs. Higher ALDH1A1 expression was observed in PD-L1+/HTIL tumors in comparison to PD-L1−/LTIL carcinomas (85% vs. 37.1%; *p* = 0.001) (Figure 4).

### 3.3. Relationship between ALDH1A1, PD-L1 and TILs with Clinicopathological Characteristics and Survival in Breast Cancer Patients

#### 3.3.1. Relationship between ALDH1A1 and Clinicopathological Characteristics and Survival in Breast Cancer Patients

There were no significant differences in age, obesity, hormonal status, histology, histopathological grade, Ki67, AR expression, positive lymph node involvement, tumor subtype (TN or HER2+) or chemotherapy regimen received between positive and negative ALDH1A1 expression. Nevertheless, positive ALDH1A expression was related to a smaller tumor size (≤5 cm) (*p* = 0.018) (Table 2). On the other hand, cases with positive ALDH1A1 expression achieved a higher pCR rate in comparison to negative ALDH1A1 carcinomas (60% vs. 37.1%; *p* = 0.048). However, there were no differences when this was analyzed according to BC subtypes. Additionally, there were no differences in DFS (*p*-log rank = 0.176) and OS (*p*-log rank = 0.487) between patients with positive and negative ALDH1A1 expression.

#### 3.3.2. Relationship between PD-L1 and Clinicopathological Characteristics and Survival in Breast Cancer Patients

There were no significant differences in age, obesity, hormonal status, histology, histopathological grade, Ki67, AR expression, positive lymph node involvement, tumor subtype (TN or HER2+), chemotherapy regimen received or pCR rate between positive and negative PD-L1 expression. Similarly, to positive ALDH1A1 tumors, smaller carcinomas (≤5 cm) more frequently showed positive PD-L1 staining compared to larger tumors (*p* = 0.004) (Table 3). Moreover, there were no statistically significant differences in DFS (*p*-log rank = 0.589) or OS (*p*-log rank = 0.706).

#### 3.3.3. Relationship between TILs and Clinicopathological Characteristics and Survival in Breast Cancer Patients

There were no significant differences in age, obesity, histology, histopathological grade, Ki67, AR, positive lymph node involvement, tumor subtype (TN or HER2+), chemotherapy regimen received or pCR rate between cases with HTILs and LTILs. Nevertheless, HTILs were more frequent in postmenopausal patients than in premenopausal patients (*p* = 0.035). Similarly, HITL infiltration was related to a smaller tumor size (≤5 cm) (*p* < 0.001) (Table 4). Regarding clinical outcomes, at the end of the follow-up (54.4 [38.3–87.6] months), 69.7% cases with HITLs and 47% cases with LTILs remained disease-free, relating HTILs to a higher DFS (*p*-log rank test = 0.027). However, no statistically significant differences in OS were found (*p*-log rank = 0.105).

#### 3.3.4. Predictors of DFS and OS in the Cohort

In the univariate analysis the presence of HTILs (HR 0.39 95% CI (0.17–0.93); *p* = 0.033) and a tumor size > 5 cm (HR 3.95 95% CI (1.78–8.75); *p* = 0.01) were the only variables associated with a better and a worse DFS, respectively. However, in the entire cohort, a tumor size > 5 cm was the only variable in the multivariate analysis related to a worse DFS (HR 3.37 95% CI (1.30–8.76); *p* = 0.012). On the other hand, regarding the risk of death, the presence of positive AR (HR 0.37 (0.15–1.92); *p* = 0.031) and a tumor size > 5 cm (HR 6.53 (2.18–19.56); *p* = 0.001) were the only variables related to a better and worse OS, respectively. This remained statistically significant when both variables were included in the multivariate analysis (tumor size > 5 cm: HR 8.12 95% CI (2.24–29.45); *p* = 0.001, positive AR: HR 0.32 95% CI (0.13–0.80); *p* = 0.014). Neither ALDH1A1, PD-L1 or HTILs were related to a better OS in this analysis (Table 5).

## 4. Discussion

To the best of our knowledge, this is the first study reporting the relationship among ALDH1A1 expression, PD-L1 expression and TIL infiltration in BC patients. The association of these three biomarkers with clinicopathological characteristics, pathologic response after neoadjuvant chemotherapy and prognosis in this setting was analyzed. The main findings of the present study were as follows: (1) positive ALDH1A1 expression was related to positive PD-L1 expression and HTILs, (2) pathologic complete response was more frequently achieved by patients with positive ALDH1A1 carcinomas, (3) smaller tumors more frequently showed positive ALDH1A1 and PD-L1 expression and HTILs, and (4) patients with HTILs have better DFS than LTIL cases.

In the search for new therapeutic lines in the immunotherapy era, the study of the tumor microenvironment and the interconnexion of cells present in it has become particularly relevant. Furthermore, different factors related to relapse in BC patients have been suggested. CSC hypothesis proposed the existence of a group of cells with the abilities of differentiation, self-renewal and proliferation. Moreover, the resistance of these cell population to treatments would explain the development of metastasis and relapse in BC patients. ALDH is a surface marker used to identify CSC in multiples tumors such as BC, and ALDH1A1 is one of the most investigated isotypes. Ginestier et al. demonstrated, for the first time, increased tumorigenic activity in positive ALDH1A1 BC cells and its association with worst clinical outcomes [3]. Few studies have reported a correlation between positive ALDH1A1 expression and lower pathologic response in BC [6,14,15]

On the other hand, CSC are able to evade immune surveillance through interaction with immune cells in the tumor niche. These primary cells inhibit CD8+ lymphocyte proliferation and activate the release of cytokines using CD4+ lymphocytes that leads to perpetuate their existence [16], which will be the origin of the metastasis and relapse. Other immune escape mechanisms used by cancer cells to evade the host immune system are immune check points such as the PD-1/PD-L1 axis. Binding PD-L1 to PD-1 induces exhaustion or apoptosis of TILs that suppresses their response. PD-L1 expression has been proposed as a molecular shield on cancer cells protecting them from lysis by cytotoxic lymphocytes [17]. Moreover, in different organ systems, it has been observed that ALDH expression enhances retinoic acid production in multiple cell types modulating regulatory T cell activity [18].

In cellular models of lung cancer and melanoma, ALDH1A3 and PD-L1 expression are correlated. However, the ALDH1A1 isotype was not related to PD-L1 expression. This could indicate that different isoforms of ALDH1 can play different rolls modulating the response of the immune system [19]. In addition, Masciale et al. identified a correlation between positive ALDH expression and CD3+ and CD8+ T lymphocytes in lung cancer; however, no correlation was found with CD4+. The authors concluded that this might be explained by the dualistic role of CD4+ T lymphocytes in antitumor response [20].

In a BC setting, Almozyan et al. found a positive association between PD-L1 expression and the expression of stemness-related genes. Additionally, this study showed, in vivo, that PD-L1 expression sustains and promotes diverse factors that play a direct role in maintaining CSC stemness. This could be mediated through PI3K/AKT activation or be independent [21]. Furthermore, in a cohort that included 440 breast invasive carcinomas, Polónia et al. found a positive correlation between positive ALDH1 expression with HTILs and positive PD-L1 expression in TN breast tumors [22]. In concordance with previously reported data, in this cohort, a positive ALD1HA1 expression was related to PD-L1 expression and HTIL infiltration, although this was observed in TN tumors but not in HER2+ carcinomas. These results support the hypothesis that in BC, ALDH1 expression modulates an antitumoral immune response through PD-L1 expression, and this might be limited to certain tumor subtypes.

There is limited data regarding the association between ALDH1A1 expression and TILs in different neoplasms. To the best of our knowledge, this is the first description of ALDH1A1 expression and its relationship with TIL subtypes. In TN cases, there was a positive correlation between positive ALH1A1 expression with higher infiltration by CD3+, CD4+ and CD8+ T lymphocytes. Additionally, positive ALDH1A1 carcinomas showed B CD20+ cell infiltration with a higher frequency than tumors with negative ALDH1A1 staining. ALDH plays a crucial role in hematopoietic stem cell differentiation through the retinoic acid pathway, which is essential during B cell development and differentiation, and antibody generation [23]. Changes in ALDH activity could modulate retinoic acid production and the differentiation of B lymphocytes. Moreover, mutations in genes related to B cell function have been described in some subtypes of TN BC [24]. This might explain the positive relationship between ALDH1 and CD20+ cells in TN carcinomas found in this cohort. More studies are needed to understand the function of ALDH1 in modulating an antitumoral immune response, the role of each lymphocyte subpopulation, and the meaning of this interconnexion on the tumor microenvironment in different BC subtypes.

Previous publications have linked positive ALDH1 staining with chemoresistance. Similarly, in BC, positive ALDH1 expression has been associated with a lower pCR rate after treatment with paclitaxel and epirubicine in the neoadjuvant setting. Additionally, in patients who did not achieve pCR, the percentage of positive ALDH1 tumor cells significantly increased after neoadjuvant chemotherapy [14]. Nevertheless, Resetkova et al. did not find changes in ALDH1 expression in tumor samples before and after neoadjuvant treatment in BC [25]. In this study, positive ALDH1 expression was related to pCR in the entire cohort. The key for this relationship might be in the BC subtypes included in this study. In BC cell lines, HER2 expression increases the positive ALDH CSC population which displays increased expression of SC regulatory genes, increased invasion in vitro and tumorigenesis in animal models. Treatment with trastuzumab blocked this effect on sensitive cell lines but not on resistant ones. Furthermore, the clinical efficacy of trastuzumab may be related to its ability to target the cancer stem cell population in HER2-amplified tumors [26]. More than half of the cases included in this cohort were HER2+, and 46.5% of these achieved pCR. A hypothesis could be that the decrease in the ALDH1 positive CSC population, secondary to treatment with trastuzumab, might cause a higher pCR rate. However, due to the borderline *p*-value, the low number of patients included in this analysis and the absence of correction by other variables, this correlation must be validated in a bigger cohort and should be interpreted with caution. In addition, the lack of differences when analyzed by tumor subtype in this cohort might be due to the scarce number of patients included in each group.

Contrary to previously reported data, in this cohort, positive ALDH1A1 tumors were smaller than those that did not exhibit ALDH1A1 expression. Furthermore, 24% of the total cases showed positive nuclear ALDH1A1 staining, in addition to cytoplasmic ALDH1A1 expression. An inverse relationship between nuclear ALDH expression and cell proliferation has been described in the cornea [27]. Moreover, in low-grade colon adenomas, ALDH1A1 expression was higher than in higher grade adenomas [28]. The loss of its expression might be expected in larger tumors that typically show increased cell rate proliferation. Curiously enough, in this study, contrary to previous data, positive PD-L1 expression and HTIL infiltration were more frequent in smaller tumors as well. Nevertheless, if ALDH1 modulates PD-L1 which regulates TIL infiltration, one would expected to find a higher biomarker expression in smaller tumors. However, the small number of patients included in this analysis should be taken into account before drawing any conclusions.

Regarding survival, the presence of HTIL infiltration was the only biomarker related to a higher DFS in the entire population in the univariate analysis. However, this relationship was not observed in the subsequent multivariant analysis. Published data concerning the association between TILs and survival in BC patients are mixed. In this line, Shenasa et al. assessed the interaction of chemotherapy with different biomarkers such as TILs in BC. These authors reported the association of stromal TILs with improved invasive DFS but failed to predict the benefit from cyclophosphamide-based adjuvant chemotherapy in the full study set [29]. Moreover, the specific role of different lymphocyte subtypes in tumor immune response remains unclear. In the study previously mentioned, the presence of high CD8+ TILs was predictive of a chemotherapy benefit in nonluminal subtypes. On the other hand, the presence of CD4+ and CD8+ T cells has been related to better DFS and OS in TN BC patients [30]. Similarly, even though CD3+ T and CD20+ B infiltration have been less documented, both subtypes have been associated with a favorable prognosis in BC patients as well [31,32]. Variations in evaluation methods, the tumor subtypes included and different tumor stages between series could explain the discrepancies in results. Additionally, for some biomarkers such as ALDH1A1 and TILs, a standardized evaluation method is still missing, while the antibodies used for PD-L1 identification depends on the type of neoplasm.

This study has several limitations, mainly due to its retrospective nature and the number of patients included. Additionally, differences that might exist between the two analyzed subtypes and the lack of comparison to luminal carcinomas should be taken into account. However, the direct relationship among ALDH1A1, PD-L1 and TILs observed in the cohort should be explored in a larger population before its translation to any clinical setting, as they may be potential targets for combined therapeutic modalities including standard chemotherapy, targeted agents and immunotherapy in specific BC subtypes. In this regard, ongoing trials are exploring combinations of different immunotherapy agents and chemotherapy regimens to enhance the host immune response in the neoadjuvant setting in triple negative and HER2-positive BC [33,34,35]. Promising results from these trials are expected. The knowledge of different biomarkers related to antitumor immunity such as ALDH1A1 that could have an impact on pathologic response in certain BC subtypes and at the same time, are related to antitumor immunity, might help to elucidate which patients are the best candidates for this type of combined therapeutic modalities. Furthermore, it is well recognized that the efficacy of treatments such as chemotherapy and immunotherapy depends on the quantity and quality of the immune-activated effector cells and its antitumoral response. Due to this, it is crucial to elucidate the relationship between immune cells present in the tumor microenvironment, tumor cells and CSC since the evidence indicates that these could be responsible for cancer initiation and metastasis.

## 5. Conclusions

Positive ALDH1A1 expression was related to positive PD-L1 staining and HTILs. The presence of these three biomarkers was more frequent in smaller tumors. Patients with positive ALDH1A1 expression achieved a higher pCR rate. Furthermore, patients with HTIL infiltration showed better DFS. Interaction among these biomarkers on the tumor microenvironment and their implications in tumor response and survival in BC patients need to be investigated in future studies.

## Figures and Tables

**Figure 1 cancers-14-04418-f001:**
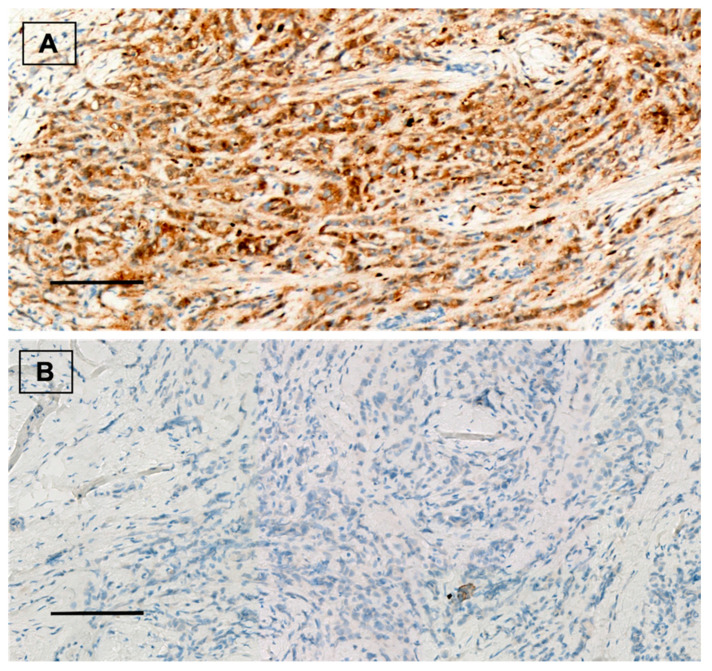
(**A**) ALDH1A1 positive staining in breast carcinoma. Immunohistochemical staining for ALDH1A1 (brown) with Mayer’s hematoxylin counterstain. ×20, scale bar = 100 µm. (**B**) ALDH1A1 negative control. ×20, scale bar = 100 µm.

**Figure 2 cancers-14-04418-f002:**
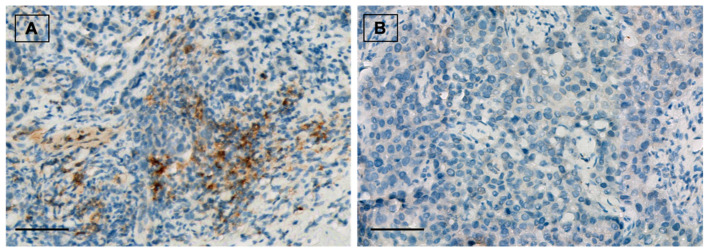
(**A**) PD-L1 positive staining in breast carcinoma. Immunohistochemical staining for PD-L1 (brown) with Mayer’s hematoxylin counterstain. ×20, scale bar = 100 µm. (**B**) PD-L1 negative control. ×20, scale bar = 100 µm.

**Figure 3 cancers-14-04418-f003:**
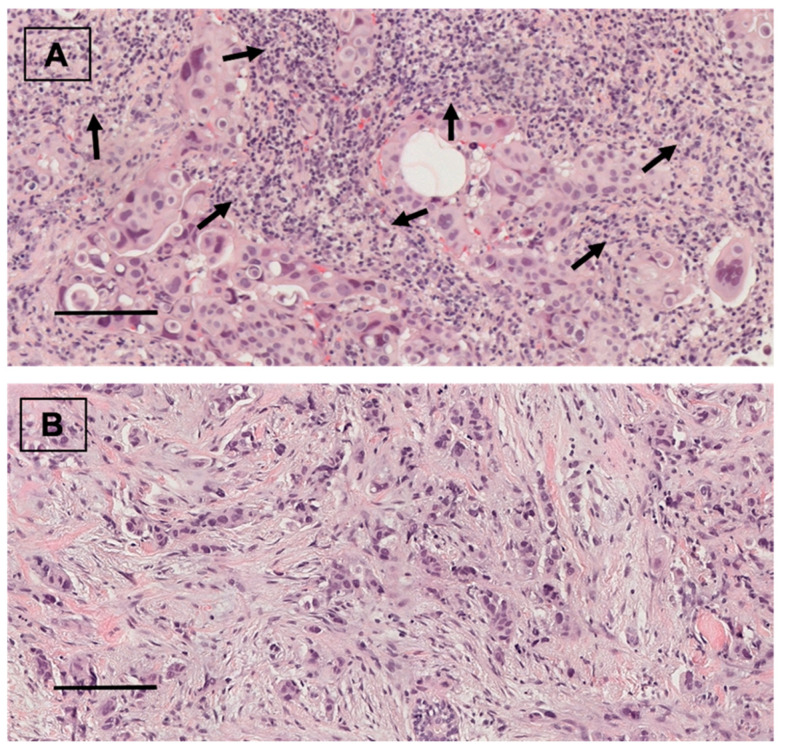
(**A**) TIL infiltration in breast carcinoma (arrows). Hematoxylin and eosin staining. ×20. scale bar = 100 µm. (**B**) Breast carcinoma without TIL infiltration. Hematoxylin and eosin staining. ×20, scale bar = 100 µm.

**Figure 4 cancers-14-04418-f004:**
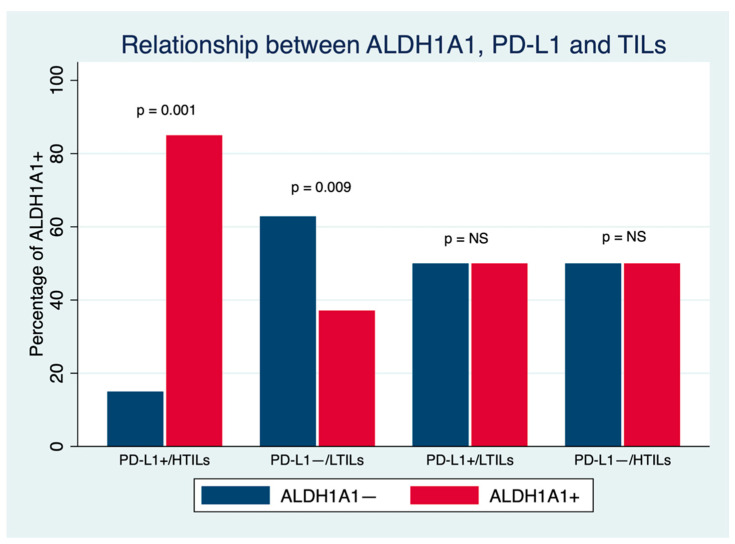
Relationship between ALDH1A1, PD-L1 and TILs in breast cancer. NS: Nonsignificant.

**Table 1 cancers-14-04418-t001:** Clinicopathologic characteristics of the entire cohort.

Variables	*n* = 75
Age	53.6 ±11.7 years
Obesity	21 (28%)
Menopausal status	
- Premenopausal	34 (45.3%)
- Postmenopausal	41 (54.7%)
Histology type	
- Ductal	67 (89.3%)
- Lobular	6 (8%)
- Metaplastic	2 (2.7%)
Histopathological grade	
- Well differentiated	3 (4%)
- Moderately differentiated	23 (30.7%)
- Poorly differentiated	49 (65.3%)
Ki67	50% (35–80%)
Positive androgen receptor	43 (57.33%)
Tumor size	
- >5 cm	32 (42.67%)
- ≤5 cm	43 (57.33%)
Positive lymph nodes	52 (69.33%)
Tumor subtype	
- HER2+ ^(a)^	43 (57.3%)
- TN ^(b)^	32 (42.7%)
Neoadjuvant therapy	
- Anthracycline	74 (98.6%)
- Taxanes	74 (98.6%)
- Carboplatin	2 (6.25%)
- Anti-HER2 therapy	39 (52%)
Pathologic complete response	37 (49.33%)
ALDH1A1 ^(c)^ expression	
- Positive	40 (53%)
- Negative	35 (47%)
PD-L1 ^(d)^ expression	
- Positive	28 (37.3%)
- Negative	47 (62.7%)
TILs ^(e)^	
- HTILs ^(f)^	32 (42.7%)
- LTILs ^(g)^	43 (57.3%)

(a) HER2+: human epidermal growth factor receptor 2-positive; (b) TN: triple negative; (c) ALDH1A1: aldehyde dehydrogenase; (d) PD-L1: programmed death ligand 1; (e) TILs: tumor-infiltrating lymphocytes; (f) HTILs: high tumor-infiltrating lymphocytes; (g) LTILS: low tumor-infiltrating lymphocytes.

**Table 2 cancers-14-04418-t002:** Relationship between ALDH1A1 expression and clinicopathological characteristics in breast cancer patients.

Variables	ALDH1+ (*n* = 40) ^(a)^	ALDH1− (*n* = 35) ^(b)^	*p* Value
Age	53.8 (±12)	53.1 (±11)	0.805
Obesity	13 (32.5%)	8 (22.9%)	0.353
Menopausal status			
- Premenopausal	14 (35.0%)	20 (57.1%)	0.055
- Postmenopausal	26 (65.0%)	15 (42.9%)	
Histology type			
- Ductal	35 (87.5%)	32 (91.4%)	0.087
- Lobular	5 (12.5%)	1 (2.9%)	
- Metaplastic	0 (0%)	2 (5.7%)	
Histopathological grade			
- Well differentiated	1 (2.5%)	2 (5.7%)	0.587
- Moderately differentiated	14 (35.0%)	9 (25.7%)	
- Poorly differentiated	25 (62.5%)	24 68.6%)	
Ki67	50% (32.5–80%)	50% (30–70%)	0.390
Positive androgen receptor	22 (55.0%)	21 (60.0%)	0.662
Tumor size			
- >5 cm	12 (30.0%)	20 (57.1%)	0.018
- ≤5 cm	28 (70.0%)	15 (42.9%)	
Positive lymph nodes	26 (65%)	26 (74.3)	0.384
- Tumor subtype			
- HER2+ ^(c)^	23 (57.5%)	20 (57.1%)	0.975
- TN ^(d)^	17 (42.5%)	15 (42.9%)	
Neoadjuvant therapy			
- Anthracycline	39 (97.5%)	35 (100%)	1.000
- Taxanes	40 (100%)	34 (97.1%)	0.467
- Carboplatin	2 (5%)	0 (0%)	0.495
- Anti-HER2 therapy	22 (55.0%)	17 (48.6%)	0.578
Pathologic complete response			
- Yes	24 (60%)	13 (37.14%)	0.048
- No	16 (40%)	22 (62.86%)	

(a) ALDH1+: aldehyde dehydrogenase positive; (b) ALDH1-: aldehyde dehydrogenase negative; (c) HER2+: human epidermal growth factor receptor 2-positive; (d) TN: triple negative.

**Table 3 cancers-14-04418-t003:** Relationship between PD-L1 expression and clinicopathological characteristics in breast cancer patients.

Variables	PDL1+ (*n* = 28) ^(a)^	PDL1− (*n* = 47) ^(b)^	*p* Value
Age	52.4 (±10.4)	54,1 (±12.4)	0.547
Obesity	7 (25%)	14 (29.8%)	0.655
Menopausal status			
- Premenopausal	13 (46.4%)	21 (44.8%)	0.883
- Postmenopausal	15 (53.4%)	26 (55.3%)	
Histology type			
- Ductal	26 (92.9%)	41 (87.2%)	0.481
- Lobular	1 (3.6%)	5 (10.6%)	
- Metaplastic	1 (3.6%)	1 (2.13%)	
Histopathological grade			
- Well differentiated	2 (7.1%)	1 (2.1%)	0.380
- Moderately differentiated	10 (35.7%)	13 (27.7%)	
- Poorly differentiated	16 (57.1%)	33 (70.2%)	
Ki67	50% (40–80%)	50% (30–80%)	0.154
Positive androgen receptor	16 (37.2%)	27 (62.7%)	0.979
Tumor size			
- >5 cm	6 (21.4%)	26 (55.3%)	0.004
- ≤5 cm	22 (78.6%)	21 (44.7%)	
Positive lymph nodes	21 (75%)	31 (66%)	0.411
Tumor subtype			
- HER2+ ^(c)^	15 (53.6%)	28 (59.6%)	0.611
- TN ^(d)^	13 (46.4%)	19 (40.4%)	
Neoadjuvant therapy			
- Anthracycline	28 (100%)	46 (97.9%)	1.000
- Taxanes	28 (100%)	46 (97.9%)	1.000
- Carboplatin	0 (0%)	2 (4.3%)	0.526
- Anti-HER2 therapy	14 (50%)	25 (53.2%)	0.789
Pathologic complete response			
- Yes	15 (53.6%)	22 (46.8%)	0.571
- No	13 (46.4%)	25 (53.2%)	

(a) PD-L1+: programmed death ligand 1 positive; (b) PD-L1−: programmed death ligand 1 negative; (c) HER2+: human epidermal growth factor receptor 2-positive; (d) TN: triple negative.

**Table 4 cancers-14-04418-t004:** Relationship between TILs and clinicopathological characteristics in breast cancer patients.

Variables	HTILs (*n* = 32) ^(a)^	LTILs (*n* = 43) ^(b)^	*p* Value
Age	54.1 (±9.7)	52.9 (±13)	0.663
Obesity	12 (37.5%)	9 (20.9%)	0.114
Menopausal status			
- Premenopausal	10 (31.2%)	24 (55.8%)	0.035
- Postmenopausal	22 (68.8%)	19 (44.2%)	
Histology type			
- Ductal	30 (93.8%)	37 (86.1%)	0.597
- Lobular	2 (6.3%)	4 (9.3%)	
- Metaplastic	0 (0%)	2 (4.7%)	
Histopathological grade			
- Well differentiated	1 (3.1%)	2 (4.7%)	0.584
- Moderately differentiated	12 (37.5%)	11 (25.6%)	
- Poorly differentiated	19 (59.4%)	30 (69.7%)	
Ki 67	50% (32.5–80%)	50% (30–80%)	0.667
Positive androgen receptor	18 (56.3)	25 (58.1)	0.870
Tumor size			
- >5 cm	6 (18.8%)	26 (60.5%)	<0.001
- ≤5 cm	26 (81.2%)	17 (39.5%)	
Positive lymph nodes	19 (59.4%)	33 (76.7%)	0.107
Tumor subtype			
- HER2+ ^(c)^	19 (59.4%)	24 (55.8%)	0.758
- TN ^(d)^	13 (40.6%)	19 (44.2%)	
Neoadjuvant therapy			
- Anthracycline	32 (100%)	42 (97.7%)	1.000
- Taxanes	32 (100%)	42 (97.7%)	1.000
- Carboplatin	1 (3.1%)	1 (2.3%)	1.000
- Anti-HER2 therapy	17 (53.1%)	22 (51.2%)	0.866
- Pathologic complete response			
- Yes	18 (56.3%)	19 (44.2%)	0.301
- No	14 (43.7%)	24 (55.8%)	

(a) HTILs: high tumor-infiltrating lymphocytes; (b) LTILs: low tumor-infiltrating lymphocytes; (c) HER2+: human epidermal growth factor receptor 2-positive; (d) TN: triple negative.

**Table 5 cancers-14-04418-t005:** Predictors of disease-free and overall survival in breast cancer patients.

Variables	DFS ^(a)^		OS ^(b)^	
	Univariante Analysis	Multivariante Analysis	Univariante Analysis	Multivariante Analysis
	HR (95% CI); *p* Value	HR (95% CI); *p* Value	HR (95% CI); *p* Value	HR (95% CI); *p* Value
Age	1.0 (0.97–1.03);		1.01 (0.97–1.05);	
	*p* = 0.939		*p* = 0.662	
Obesity	0.69 (0.28–1.70);		0.98 (0.89–1.09);	
	*p* = 0.416		*p* = 0.748	
Hormonal status	0.99 (0.47–2.10);		1.54 (0.62–3.87);	
	*p* = 0.995		*p* = 0.355	
Histology type (against ductal type) − Lobular − Metaplastic	1.40 (0.42–4.66)		1.38 (0.32–5.99)	
	7.51 (1.66–34.04);		7.42 (1.67–32.90);	
	*p* = 0.105		*p* = 0.110	
TN ^(c)^ (against HER2+) ^(d)^	1.37 (0.65–2.89);		2.02 (0.83–4.88);	
	*p* = 0.403		*p* = 0.120	
Ki67	1.00 (0.99–1.02);		1.02 (0.99–1.03);	
	*p* = 0.796		*p* = 0.214	
Positive androgen receptor	0.62 (0.29–1.30);		0.37 (0.15–1.92);	0.32 (0.13–0.80);
	*p* = 0.201		*p* = 0.031	*p* = 0.014
Tumor size > 5 cm	3.95 (1.78–8.75);	3.37 (1.30–8.76);	6.53 (2.18–19.56);	8.12 (2.24–29.45);
	*p* = 0.01	*p* = 0.012	*p* = 0.001	*p* = 0.001
Positive lymph nodes	2.74 (0.95–7.91);		1.75 (0.59–5.24);	
	*p* = 0.062		*p* = 0.316	
ALDH1A1+ ^(e)^	0.060 (0.28–1.27);		0.073 (0.30–1.77);	
	*p* = 0.181		*p* = 0.489	
PD-L1+ ^(f)^	0.81 (0.36–1.79);		0.83 (0.32–2.17);	
	*p* = 0.598		*p* = 0.707	
HTILs ^(g)^	0.39 (0.17–0.93);	0.68 (0.26–1.75);	0.44 (0.16–0.22);	
	*p* = 0.033	*p* = 0.424	*p* = 0.115	

(a) DFS: disease-free survival; (b) OS: overall survival; (c) TN: triple negative, (d) HER2+: human epidermal growth factor receptor 2-positive; (e) ALDH1A1+: positive aldehyde dehydrogenase; (f) PD-L1+: positive programmed death ligand 1; (g) HTILs: high tumor-infiltrating lymphocytes.

## Data Availability

The data that support the findings of this study are available for the Oncology Investigation Unit of the University Hospital of León, but restrictions apply to the availability of these data. Data are, however, available from the authors upon reasonable request and with permission of the local Ethics Committee of the University Hospital of León.

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
