# Peer review of "Relationship between Aldehyde Dehydrogenase, PD-L1 and Tumor-Infiltrating Lymphocytes with Pathologic Response and Survival in Breast Cancer"

_cancers, 2022, doi:10.3390/cancers14184418_

Round 1

Reviewer 1 Report

The manuscript from Lopez Flores et al. describes the relationship between expression of three different markers used to characterise breast cancers samples, ALDH1A1, programmed death ligand 1 (PD-L1) and tumor-infiltrating lymphocytes (TIL). The authors focused on triple-negative and HER2+ breast cancer specimens, and they extended their investigation to specific sub-populations of lymphocytes, namely CD3+, CD4+, CD8+ and CD20+ cells. They established then associations to clinico-pathological characteristics, response to neoadjuvant therapy and clinical outcomes. Lopez Flores et al. found positive correlation between expression of ALDH1A, PD-L1 and TIL in these tumours. They also observed that expression of these markers was statistically more common in tumours smaller than 5cm and that ALDH1A + tumours were more likely to show pathological complete response (pCR). Big tumour size was also associated to reduced disease-free and overall survival following multivariate statistical analysis.

The manuscript shows that expression of cancer-stem cell marker, regulator of tumour immune response and presence of immune infiltrate are correlated in these subtypes of breast cancers, as previously suggested in literature, and they can be more commonly represented in smaller tumours. The authors discussed the interesting and debated idea that reduced tumour cell proliferation can result in over-representation of stem cell markers and that this may in turn lead to up-regulation of mechanisms of immune escape and increased infiltration of immune cells.

The study is overall well conducted, it adds to the general knowledge of relationships between markers of cancer stem cells and immune response, while it offers interesting hypothesis for further studies. However, the following issues should be addressed:

1.    The relationship between pCR and ALDH1A positivity is not very solid, being the p value of the association just below 0.05, while no correction for multiple statistical testing was implemented in the analysis. The authors should reiterate in the discussion that this correlation should be validated in a bigger cohort of samples and that this association should be interpreted with caution. 

2.    Figures 1, 2 and 3, showing immunohistochemistry staining examples, cannot be interpreted without a negative control. A negative control demonstrating specific interaction between antibody and epitope (e.g. isotype control) should be added to the figures.

3.    Table 2 reports numbers and percentages of patients in the tumour size section that don’t support the conclusions made by the authors. Please verify that numbers and headings are correct.  

Other minor points are:

1.    Figure 4 should report headings written in English

2.    Reference n. 3 in the introduction shouldn’t be referred to the role of ALDH1A in oxidation of retinol to retinoic acid during stem cell differentiation. An example of more appropriate reference could be PMID: 19806016 Ginestier et al. Cell Cycle 2009.

Author Response

POINT-BY-POINT: RESPONSE TO THE REVIEWERS

Thank you for the opportunity to answer the questions raised by the reviewers. Their suggestions will undoubtedly improve the manuscript. Here, we answer point-by-point all the comments:

Reviewer #1:

The manuscript from Lopez Flores et al. describes the relationship between expression of three different markers used to characterise breast cancers samples, ALDH1A1, programmed death ligand 1 (PD-L1) and tumor-infiltrating lymphocytes (TIL). The authors focused on triple-negative and HER2+ breast cancer specimens, and they extended their investigation to specific sub-populations of lymphocytes, namely CD3+, CD4+, CD8+ and CD20+ cells. They established then associations to clinico-pathological characteristics, response to neoadjuvant therapy and clinical outcomes. Lopez Flores et al. found positive correlation between expression of ALDH1A, PD-L1 and TIL in these tumours. They also observed that expression of these markers was statistically more common in tumours smaller than 5cm and that ALDH1A + tumours were more likely to show pathological complete response (pCR). Big tumour size was also associated to reduced disease-free and overall survival following multivariate statistical analysis.

The manuscript shows that expression of cancer-stem cell marker, regulator of tumour immune response and presence of immune infiltrate are correlated in these subtypes of breast cancers, as previously suggested in literature, and they can be more commonly represented in smaller tumours. The authors discussed the interesting and debated idea that reduced tumour cell proliferation can result in over-representation of stem cell markers and that this may in turn lead to up-regulation of mechanisms of immune escape and increased infiltration of immune cells.

The study is overall well conducted, it adds to the general knowledge of relationships between markers of cancer stem cells and immune response, while it offers interesting hypothesis for further studies. However, the following issues should be addressed:

  1. The relationship between pCR and ALDH1A positivity is not very solid, being the p value of the association just below 0.05, while no correction for multiple statistical testing was implemented in the analysis. The authors should reiterate in the discussion that this correlation should be validated in a bigger cohort of samples and that this association should be interpreted with caution.

Response: Thank you for your comment. As you correctly pointed out, due to the number of patients included in this analysis the association between pCR and ALDH1A1 positivity found in this study should be validated in a bigger cohort of samples and this association should be interpreted with caution. This has been added in the discussion as a limitation.

  1. Figures 1, 2 and 3, showing immunohistochemistry staining examples, cannot be interpreted without a negative control. A negative control demonstrating specific interaction between antibody and epitope (e.g. isotype control) should be added to the figures.

Response: Thank you for your comment. Images of negative control have been added to figures 1, 2 and 3.

  1. Table 2 reports numbers and percentages of patients in the tumour size section that don’t support the conclusions made by the authors. Please verify that numbers and headings are correct.

Response: Thank you for your comment. This table has been verified and corrected in the text.

Other minor points are:

  1. Figure 4 should report headings written in English

Response: Thank you for your comment. Figure 4 written in English have been added in the text.

  1. Reference n. 3 in the introduction shouldn’t be referred to the role of ALDH1A in oxidation of retinol to retinoic acid during stem cell differentiation. An example of more appropriate reference could be PMID: 19806016 Ginestier et al. Cell Cycle 2009.

Response: Thank you for your thoughtful suggestion. The next sentence has been added to the introduction: ¨Moreover, ALDH1A1 seems to play a role in the early differentiation of breast cancer stem cells¨. The appropriated reference has been added in the text.

Reviewer 2 Report

The current manuscript under review attempts to show the relationship between ALDH1A1, PD-L1 expression, and TILs infiltration in BC patients. These three biomarkers were analyzed together with clinicopathological characteristics, pathologic response after neoadjuvant chemotherapy, and prognosis. Some of the more prevalent recent studies for biomarker expression have focused on building/testing more robust statistical models, including a larger dataset compared to the current study. The manuscript reports no statistical differences were observed in most of the analyzed clinicopathological characteristics except for tumor size. The study also lacks a clear direction on how these biomarkers would be relevant for their translation to a clinical setting. The manuscript in its current form does not meet the standards of the journal.

The authors are requested to evaluate the following recent clinical study to add more relevant information. Shenasa E, Stovgaard ES, Jensen M-B, Asleh K, Riaz N, Gao D, Leung S, Ejlertsen B, Laenkholm A-V, Nielsen TO. Neither Tumor-Infiltrating Lymphocytes nor Cytotoxic T Cells Predict Enhanced Benefit from Chemotherapy in the DBCG77B Phase III Clinical Trial. Cancers. 2022; 14(15):3808. https://doi.org/10.3390/cancers14153808

Additional comments for the authors:

For a positive immunohistology stain, it is mentioned that a stain is positive if more than 1% of tumor of cells were expressed the marker of interest. Were the cells calculated manually? The authors are requested to provide a detailed methodology.

In the results section, all the histological images for figures 1-3 show the positive staining but requires a complementary negative stain image for comparison. Additionally, please provide scale bars for the images.

In figure 4, the authors are advised to translate the axis and chart titles to English.

Please proofread for typography and grammatical errors. Here is a non-exhaustive list, Line 175 – “Hight” tumor-infiltrating lymphocytes; line 255 – “stanning”.

Author Response

POINT-BY-POINT: RESPONSE TO THE REVIEWERS

Thank you for the opportunity to answer the questions raised by the reviewers. Their suggestions will undoubtedly improve the manuscript. Here, we answer point-by-point all the comments:

Reviewer #2:

The current manuscript under review attempts to show the relationship between ALDH1A1, PD-L1 expression, and TILs infiltration in BC patients. These three biomarkers were analyzed together with clinicopathological characteristics, pathologic response after neoadjuvant chemotherapy, and prognosis. Some of the more prevalent recent studies for biomarker expression have focused on building/testing more robust statistical models, including a larger dataset compared to the current study. The manuscript reports no statistical differences were observed in most of the analyzed clinicopathological characteristics except for tumor size. The study also lacks a clear direction on how these biomarkers would be relevant for their translation to a clinical setting. The manuscript in its current form does not meet the standards of the journal.

Response: Thank you for your comment. As you described no statistical differences were observed in most of the analyzed clinicopathological characteristics except for tumor size. The number of patients included in this analysis might explain these results. This has been added in the main text as a study limitation that should be taken into account when interpreting the results. Regarding on how these biomarkers would be relevant for their translation to a clinical setting has now been reported in the main text (discussion section):

¨ This study has several limitations, mainly due to its retrospective nature and the number of patients included. Additionally, differences that might exist between the two analyzed subtypes, and the lack of comparison with luminal carcinomas should be taken into account. However, the direct relationship among ALDH1A1, PD-L1 and TILs observed in the cohort should be explored in a larger population before its translation to any clinical setting, as they may be potential targets for combine therapeutic modalities including standard chemotherapy, targeted agents and immunotherapy in specifics BC subtypes. In this regard, ongoing trial are exploring combinations of different immunotherapy agents and chemotherapy regimens to enhance the host immune response in the neoadjuvant setting in triple negative and HER2 positive BC [33] [34] [35]. Promising results from these trials are expected. The knowledge of different biomarkers related to antitumor immunity such as ALDH1A1 that could have an impact on pathologic response in certain BC subtypes, and at the same time are related to antitumor immunity, might help to elucidate which patients are the best candidates for this type of combine therapeutic modalities. ¨

The authors are requested to evaluate the following recent clinical study to add more relevant information. Shenasa E, Stovgaard ES, Jensen M-B, Asleh K, Riaz N, Gao D, Leung S, Ejlertsen B, Laenkholm A-V, Nielsen TO. Neither Tumor-Infiltrating Lymphocytes nor Cytotoxic T Cells Predict Enhanced Benefit from Chemotherapy in the DBCG77B Phase III Clinical Trial. Cancers. 2022; 14(15):3808. https://doi.org/10.3390/cancers14153808

Response: Thank you for your kind suggestion that will improve the quality of the manuscript. Information regarding the results of this study has been added to the discussion:

¨Regarding to survival, the presence of HTILs infiltration was the only biomarker related to a higher DFS in the entire population in the univariate analysis. However, this relationship was not observed in the subsequent multivariante analysis. Publish date concerning the association between TILs and survival in BC patients are mixed. In this line, Shenasa et al. assessed the interaction of chemotherapy with different biomarkers such as TILs in BC. These authors reported association of stromal TILs with an improved invasive DFS, but failed to predict benefit from cyclophosphamide-based adjuvant chemotherapy in the full study set [29]. Moreover, the specific role of different lymphocytes subtypes in tumor immune response remains unclear. In the study mention previously, the presence of high CD8+ TILs was predictive of chemotherapy benefit in non-luminal subtypes¨.

Additional comments for the authors:

For a positive immunohistology stain, it is mentioned that a stain is positive if more than 1% of tumor of cells were expressed the marker of interest. Were the cells calculated manually? The authors are requested to provide a detailed methodology.

Response: Thank you for your comment. The percentage of positive cells was calculated manually in a whole section at high magnification (200×). The detailed methodology has been added to the main text as follows in the next sentence, in the methods section: ¨Afterwards, evaluation of ALDH1A1 immunohistochemistry was performed by scanning the stained slides at medium (20×) and high magnification (100×). The percentage of tumor area with positive cytoplasmic staining was estimated manually by an expert pathologist. Specimens were considered ALDH1A1 positive when the staining was more than 1 % in tumor cells¨.

In the results section, all the histological images for figures 1-3 show the positive staining but requires a complementary negative stain image for comparison. Additionally, please provide scale bars for the images.

Response: Thank you so much for your patient and careful check and thoughtful suggestions. Complementary negative stain images and scale bars have been added to figures 1, 2 and 3.

In figure 4, the authors are advised to translate the axis and chart titles to English.

Response: Thank you for your comment. This figure has been properly translated to English.

Please proofread for typography and grammatical errors. Here is a non-exhaustive list, Line 175 – “Hight” tumor-infiltrating lymphocytes; line 255 – “stanning”.

Response: Thank you for your comment. The entire manuscript has been checked for typography and grammatical errors.

Round 2

Reviewer 2 Report

The authors have addressed all my comments to satisfaction and the manuscript has been improved considerably by citing relevant previous papers. 

One suggestion to the authors would be to label the TILs in Figure 3 so that it can be easily visualized by readers. 

Author Response

POINT-BY-POINT: RESPONSE TO THE REVIEWERS

Thank you for the opportunity to answer the questions raised by the reviewers. Their suggestions will undoubtedly improve the manuscript. Here, we answer point-by-point all the comments:

Reviewer #2:

The authors have addressed all my comments to satisfaction and the manuscript has been improved considerably by citing relevant previous papers. 

One suggestion to the authors would be to label the TILs in Figure 3 so that it can be easily visualized by readers.

Response: Thank you so much for your careful check and thoughtful suggestion. TILs have been labelled with black arrows in Figure 3.